# Extreme heat and mortality in the State of Rio de Janeiro in November 2023: attribution to climate change and ENSO

Soledad Collazo[1,2,3], David Barriopedro[2], Ricardo García-Herrera[1,2], Santiago Beguería[4]

[1]Departamento de Física de la Tierra y Astrofísica, Facultad de Ciencias Físicas, Universidad Complutense de Madrid (UCM), Plaza de las Ciencias 1, Madrid, 28040, Spain
[2]Instituto de Geociencias (IGEO), Consejo Superior de Investigaciones Científicas–Universidad Complutense de Madrid (CSIC–UCM), Madrid, Spain
[3]Departamento de Ciencias de la Atmósfera y los Océanos, Facultad de Ciencias Exactas y Naturales, Universidad de Buenos Aires (FCEN, UBA), Buenos Aires, Argentina
[4]Estación Experimental de Aula Dei–Consejo Superior de Investigaciones Científicas (EEAD–CSIC), Zaragoza, Spain

*Correspondence to*: Soledad Collazo (scollazo@ucm.es)

**Abstract**. During 2023, the State of Rio de Janeiro experienced unprecedented maximum temperatures, resulting in a substantial increase in human mortality. This study aims to analyze the contribution of global warming to changes in the distribution of annual maximum temperatures and their subsequent impact on mortality rates. Our analysis of extreme temperatures reveals that a non-stationary model, in which the location parameter shifts linearly as a function of global warming and/or El Niño-Southern Oscillation (ENSO), provides a significantly better fit to the data than its stationary counterpart. The northern region of the State exhibited the strongest response to climate change, while ENSO effects were most pronounced in the eastern region. Events as likely as the 2023 record were estimated about 2°C colder in pre-industrial times. Under a 2°C global warming scenario, the probability of experiencing maximum temperatures equal to the 2023 increases by at least a factor of three. These findings highlight climate change as the primary driver of extreme temperature intensification, with ENSO acting as a secondary but significant factor in the eastern region. As global warming approaches 2°C, Rio de Janeiro is projected to experience heatwaves of that magnitude every four years approximately. Climate change has contributed to one in three heat-related deaths recorded during the peak of the event. Without adaptation and mitigation measures, global warming would further increase the death toll during extreme events of the same frequency to those experienced in 2023.

## 1 Introduction

From early austral spring 2023 to late summer March 2024, central and southern Brazil experienced extremely high daily maximum air temperatures (TX). This period recorded the warmest spring in at least 63 years for the region, with TX locally exceeding 43°C, which was 5-8°C higher than the 1991-2020 climatology (Kew et al., 2023; Perkins-Kirkpatrick et al.,

2024). The intense heat persisted throughout the season and peaked in November, when TX anomalies reached +9°C in some areas of southern Brazil. Finally, from 15 to 18 March 2024, the region again registered another exceptional heatwave, with temperatures climbing to unprecedented levels (~42°C in Rio de Janeiro) for early autumn (Faranda & Alberti, 2024).

This unusual season was accompanied by El Niño, the warm phase of the El Niño-Southern Oscillation (ENSO), which modulates the temperature and precipitation of tropical South America (Cai et al., 2020). During El Niño events, the descending branch of the Walker circulation shifts toward the tropical Atlantic Ocean and northeastern South America, encompassing the eastern Amazon region and northeastern Brazil (Reboita et al., 2021). In addition to modifications in the Walker circulation, ENSO-related impacts over South America are also modulated by tropical-extratropical teleconnections.

This mechanism involves stationary Rossby wave trains, initiated by anomalous convection over the tropical Pacific, which propagate into the mid-latitudes, generating alternating centres of high and low atmospheric pressure (Cai et al., 2020). Nevertheless, the influence of El Niño on precipitation in the state of Rio de Janeiro is weak (de Oliveira-Júnior et al., 2018; Sobral et al., 2019). In terms of temperature, this region experiences discernible warm anomalies during El Niño phase throughout the year, except for the austral winter (Cai et al., 2020). In particular, in the city of Rio de Janeiro, TX is ~1°C

warmer than the climatology during intense El Niño events, although this increase is not statistically significant (Wanderley et al., 2019).

  Besides the influence of El Niño, the extreme temperature conditions of 2023 occurred in the context of an increasingly warming planet (IPCC, 2023, Summary for Policymakers). In Brazil, de Barros Soares et al. (2017) found an overall warming, with observed near-surface air temperatures increasing by up to 1°C per decade between 1975 and 2004. Over the

ocean near the coasts of Rio de Janeiro and São Paulo, TX also shows significant annual and seasonal positive trends (de Oliveira et al., 2021). Furthermore, the frequency of occurrence of warm extremes has significantly increased over Brazil during the period 1961-2018, while the opposite is true for cold extremes (Regoto et al., 2021). It is also noteworthy that the largest increases in warm extremes occur during spring and austral summer, coinciding with the period of exceptional warmth in 2023/2024. A more detailed analysis of trends in the State of Rio de Janeiro for the period 1961-2012 reveals

significant warming in mean TX (between +0.01 and +0.08°C/year) over the metropolitan area and the northern and northwestern regions of the state, as well as upward trends in the percentage of warm nights and days (between +0.1 and +0.6 % days/year) for almost the entire state (Silva & Dereczynski, 2014). However, when the six largest cities in Brazil are considered, Rio de Janeiro shows the lowest increases in the number of heatwave days between 1961-1980 and 1981-2014, with positive but non-significant trends (Geirinhas et al., 2018).

In light of the reported changes over this region, it becomes pertinent to ask to what extent climate change contributes to the observed long-term trend in temperature and associated extremes. Using global climate models, de Abreu et al. (2019) found that anthropogenic activities account for a substantial fraction of the observed temperature trends in southeastern Brazil, with no significant contribution from natural or other sources.

  In addition to the attribution of trends in mean and extreme temperature, recent studies also focused on the attribution of

individual extreme events to climate change (NAS, 2016; Jézéquel et al., 2018; Otto, 2017; Stott et al., 2016). Attributing

individual extreme events was considered unfeasible until Allen (2003) outlined a methodology for evaluating the influence of external factors on the probability of a specific extreme weather event. Subsequently, Philip et al. (2020) described a protocol that relies heavily on statistical methods within the extreme value theory (EVT). To model the tail of the distribution, classical EVT fits a stationary probability density function (PDF) to extreme values; however, in the context of

global warming, the stationary assumption is not valid (Ouarda et al., 2020). Therefore, the protocol proposes a non-stationary approach in which the PDF shifts linearly as a function of a covariate (Katz, 2013; Robin & Ribes, 2020; Slater et al., 2021). Using global warming (a clear indicator of anthropogenic activities) as a covariate enables modeling its effect on the behavior of the tail. By applying this protocol to attribute the early spring 2023 heat in South America, Kew et al. (2023) estimated that the event would have been 1.4 to 4.3°C cooler if humans had not warmed the planet by burning fossil fuels.

Moreover, they stated that the direct contribution of ENSO to the extreme heat is small compared to the climate change signal. On the other hand, by using historical analogs of the event, Faranda & Alberti (2024) claimed that both anthropogenic climate change and natural climate variability played a role in intensifying the March 2024 heatwave in Brazil. These two rapid early- and late-season heat attribution analyses conducted in southeastern Brazil yielded varying conclusions regarding the extent to which internal variability contributes to the intensification of high temperatures. Furthermore, they employed

gridded observations and reanalyses, which may lead to an underestimation of hot extreme events due to spatial averaging (Balmaceda-Huarte et al., 2021; Sheridan et al., 2020).

Previous studies have identified that heatwaves in Brazil have severe impacts on health, particularly among vulnerable populations such as the elderly and those with pre-existing conditions. In Rio de Janeiro, prolonged heat exposure exacerbates chronic conditions and increases mortality from cardiovascular and respiratory diseases (Ferreira et al., 2019;

Silveira et al., 2023). For example, events like the 2010 heatwave led to excess mortality among older adults due to circulatory diseases (Geirinhas et al., 2020). Urban factors (heat island effect) further amplify these risks by heightening temperature anomalies in densely built-up areas (Krüger et al., 2024; Peres et al., 2018). Economic inequality also plays a critical role, as low-income populations are disproportionately vulnerable to the effects of extreme heat (Zhao et al., 2019a). Furthermore, there is little evidence of thermal adaptation at the national level, raising concerns that the health burden of

heat exposure may escalate with global warming (Zhao et al., 2019a; Zhao et al., 2019b).

In this study, we examine the contribution of climate change and ENSO to the daily maximum temperatures observed in November 2023, which were historical records over the entire period (1971-2024). The analysis is conducted at five meteorological stations in the State of Rio de Janeiro. Additionally, we assess the probability of similar extreme events occurring in the future and estimate their return periods for different global warming levels and ENSO phases. To achieve

this, we apply the EVT with a non-stationary approach (Beguería et al. 2023). This methodology allows us to account for temporal changes in the magnitude / frequency of occurrence of extreme events. Finally, we assessed the impact of the scorching temperatures recorded during this record-breaking event on mortality rates across the state, and the relative contribution of climate change to the death toll.

## 2 Data and Methods

### 2.1 Data

We used TX series from 53 weather stations in the State of Rio de Janeiro, provided by the Brazilian National Institute of Meteorology (INMET). To ensure data quality, daily minimum temperatures were also collected, enabling the replacement of TX values below the daily minimum with missing data codes. The dataset contained a substantial number of gaps. Therefore, for this study, we selected stations with less than 15% missing data from 1 January 1971 to 20 March 2024, ensuring that their coverage includes the last seven months of this period (from September 2023 to March 2024) with less than 15% missing data as well. In total, five stations met these criteria. This 15% threshold was adopted as a compromise to ensure both sufficient temporal coverage for robust analysis and broad spatial representation across the state. The selected stations are well distributed throughout the state (see Fig. 1 and Table S1), and broadly represent the diverse climate conditions of the State of Rio de Janeiro, which are spatially variable due to its complex terrain, characterized by hills, mountains, valleys, a variety of vegetation, lowlands and bays, as well as its proximity to the Atlantic Ocean (Silva et al., 2014). Special attention was given to the Itaperuna station, which exhibited the highest proportion of missing data among the five stations. Between 1983 and 1989, no TX records are available, representing the period with the highest concentration of missing data at this station (Fig. S1). More recently, during the 2023/24 season, missing values are most frequent in March.

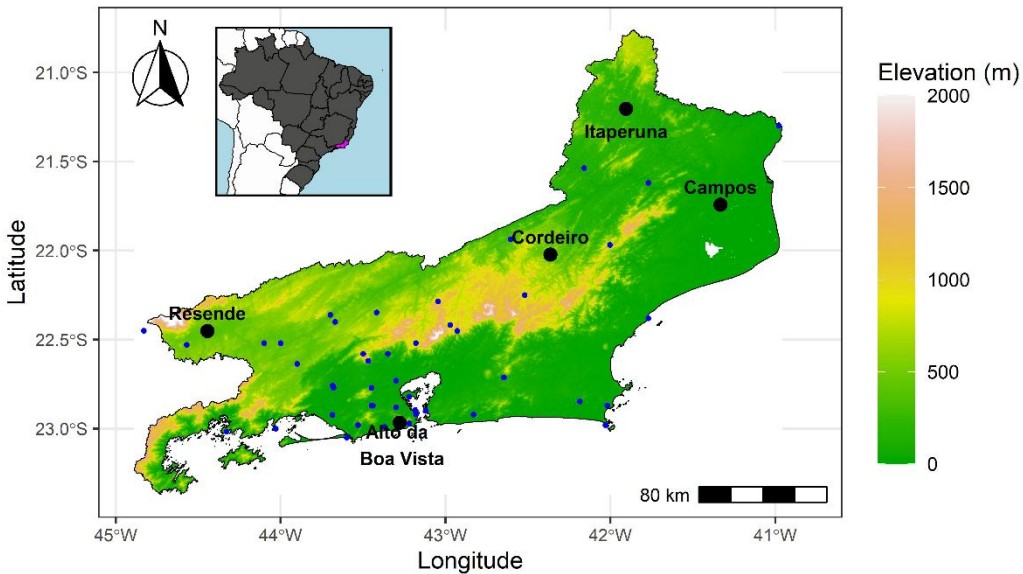

Figure 1: Weather stations in the Rio de Janeiro State. Stations with valid values in September 2023-March 2024 and less than 15% of missing data in the period 1971-2024 are identified with a black dot and the name. Elevation data were accessed via the Amazon Web Services Open Data Terrain Tiles using the elevatr R package (Hollister et al., 2023)

Once the weather stations were identified, we filled the gaps of their TX time series by using the most similar station. If both stations had missing data on the same day, the second most similar station was used. To determine station similarity, we computed the correlation of the TX series after filtering out their linear trends and annual cycles using moving averages. Next, we built the empirical cumulative distribution functions (ECDFs) of the original and similar station. Missing data were then estimated using quantiles from these distributions, a technique known as quantile mapping (Beguería et al., 2019; Devi et al., 2019; Grillakis et al., 2020). Compared to infilling by regression techniques, which tends to smooth the estimated data, quantile mapping maintains the extremes (tails) of the distribution more effectively (Beguería et al., 2019). Additionally, using ECDFs in quantile mapping helps to avoid potential biases between the two time series. Finally, we identified the extreme values of the completed TX series by using the block maxima approach. This method involves dividing the time series into non-overlapping blocks, and selecting the highest daily maximum temperature within each block. As the region under study is located within a tropical climate zone, we used annual blocks, therefore taking the TX value of the hottest day of the year (TXx). To estimate the magnitude and significance of the trend in TXx, we employed the non-parametric Sen's slope estimator (Sen, 1968) and the Mann–Kendall trend test (Kendall, 1975; Mann, 1945), respectively.

As an indicator of global warming, we used the 1850-2023 global annual mean temperature anomalies (with respect to the period 1850-1900) provided by HadCRUT5 (https://www.metoffice.gov.uk/hadobs/hadcrut5/, accessed July 2024). We applied LOESS smoothing (Cleveland et al., 1992) to this series in order to filter out interannual variability and emphasize slowly-varying anthropogenic influences. This LOESS model applies a smoothing span of 0.75, which determines the proportion of data used in each localized fit. It employs a second-degree polynomial for local regression, and assumes normally distributed errors. From this point on, we refer to these smoothed data as the Global Warming Index (GWI). This index shows anomalies close to 0°C until about 1950 and then increases rapidly to ~+1.3°C in 2023 (Fig. S2). This smoothed global mean temperature accounts for anthropogenic influence, but we cannot attribute changes to local forcings, such as aerosols, and land-use changes, which can also have large influences on extremes (Avelar & Tokarczyk, 2014; Ferreira Correa et al., 2024; Godoy et al., 2009; Solórzano et al., 2021).

To diagnose ENSO, we employed the monthly sea surface temperature (SST) anomalies in El Niño 3.4 region (5°N-5°S, 120°W-170°W), which are provided by NOAA (https://psl.noaa.gov/data/correlation/nina34.anom.data, accessed May 2025). In order to obtain an annual value, we took the SST anomalies from the month determining the TXx value of each year. This index, referred to as EN3.4 hereinafter, does not exhibit a significant linear trend and is not significantly correlated with the GWI. The smoothing applied to the GWI effectively removes potential fluctuations in global mean temperature due to ENSO. Therefore, the two covariates of TX series are independent. The sensitivity of our results to the choice of ENSO index was assessed by analysing both the Oceanic Niño Index (ONI)-calculated as the three-month running mean of sea surface temperature anomalies in the Niño 3.4 region-and the Southern Oscillation Index (SOI), using both monthly and quarterly values. For the quarterly indices, the relationship with TXx was established using the centred month of each period.

Daily mortality data for the state of Rio de Janeiro are publicly available by the Secretaria de Estado de Saude of Rio de Janeiro on its website (http://sistemas.saude.rj.gov.br/tabnetbd/dhx.exe?sim/tf_sim_do_geral.def, accessed December 2024). The period 2000-2024 was considered, but the years 2020 and 2021 were excluded from the analysis to eliminate possible disrupting effects of the pandemic of COVID-19. Moreover, for the 2023/24 season under study, additional information regarding age, sex, and cause of mortality was also collected.

## 2.2 Statistical methods

### 2.2.1 Extreme temperature attribution

To fit TXx, which represents the largest observation in a large sample (also known as block maxima), we used the generalized extreme value (GEV) distribution, as shown in Eq. (1). The choice of this statistical distribution is consistent with previous studies analyzing extreme temperatures (Coles, 2001; Van Oldenborgh et al., 2022). To assess the goodness of fit of the GEV distribution, we employed the one-sample Kolmogorov-Smirnov test (Smirnov, 1948).

$$P(x) = exp\left[-\left(1 + \xi\,\frac{x-\mu}{\sigma}\right)^{-1/\xi}\right] \tag{1}$$

The GEV distribution is characterized by three parameters: $\mu$ is the location parameter, $\sigma$ the scale parameter, and $\xi$ the shape parameter, which are related to the mean, variability, and tail behavior of the distribution, respectively. If $\xi > 0$, the distribution belongs to the Fréchet family, which has a long right tail, indicating that larger extreme events are possible and have a high probability (Eastoe, 2017). If $\xi = 0$, the GEV distribution becomes a Gumbel distribution, which models exponential tails, and has no upper or lower bound on the extremes. Finally, if $\xi < 0$, the distribution belongs to the Weibull family, which has a finite upper tail (truncated tail), implying a maximum limit for extreme values (Belzile et al., 2023). This upper bound is estimated according to Eq. (2).

$$x_{max} = \mu + \frac{\sigma}{|\xi|} \tag{2}$$

The GEV distribution expressed in Eq. (1) is stationary, meaning that its parameters remain constant over time. Therefore, to make the GEV distribution non-stationary, the parameters must be expressed as a function of one or more covariates. In the case of extreme temperatures, a reasonable and conservative hypothesis presumes simple linear relationships of the covariates with the location parameter only (Eq. (3), Kharin & Zwiers, 2005; Philip et al., 2020). Changing the location parameter $\mu$ simply shifts the distribution of extremes and changes return levels uniformly at all return periods by the same amount (Huang et al., 2016). The scaling parameter of the GEV is typically regarded as stationary when examining

temperature extremes. However, Mohammadi et al. (2024) investigated the possibility of considering this parameter to be varying exponentially with time, in conjunction with a linearly varying location parameter. The authors concluded that this

particular fit to the data was not appropriate. Regarding the shape parameter, it is a good practice to assume a stationary behavior, as otherwise it leads to large uncertainties and a failed fit of the observations (Friederichs & Hense, 2007).

In this study, we considered the stationary model ($M_{0\_S}$) and three non-stationary models where only the location parameter is linearly changing (Eq. 3). These are as follows: univariate dependent on GWI ($M_{1\_gwi}$), univariate dependent on EN3.4 ($M_{2\_en3.4}$), and multivariate dependent on GWI and EN3.4 ($M_{3\_multi}$). Considering single and combined influences of two

covariates is an approach little explored in South American attribution studies. For example, Pereira et al. (2023) applied non-stationary GEV models to extreme temperature analysis in Campinas, Brazil, using time as the sole covariate to model changes in the location parameters.

$$\mu = \beta_0 + \sum_{i=1}^{I} \beta_i Z_i + \epsilon \tag{3}$$

where $\beta_0$ is the intercept, $\beta_i$ (i = 1, …, I) are the coefficients associated with the covariates $Z_i$, and $\epsilon$ is the residual or error (Beguería et al. 2023, Collazo 2024). The parameters of the non-stationary GEV distribution, including the coefficients of the linear model in Eq. (3), were estimated simultaneously using maximum likelihood estimation. This was implemented through the ismev package in the R programming language, which provides specialized tools for fitting extreme value

models (Heffernan & Stephenson, 2018).

To test which of the models provided a better fit to the data, we performed the Likelihood Ratio Test (LRT, Coles, 2001), which compares the goodness of fit of two models based on the ratio of their likelihoods (Eq. 4).

$$D = -2\big(L(M_i) - L(M_j)\big) \tag{4}$$

where $L$ is the maximum of the log-likelihood function of the considered model. The D statistic follows a chi-squared distribution with degrees of freedom equal to the difference between the lengths of the two models. A 5% significance level is used.

As an additional model selection criterion, we also computed the Akaike Information Criterion (AIC), which considers both

the goodness of fit, which increases with the number of covariates, and a penalty factor based on the complexity of the model (Akaike, 1973; Cavanaugh & Neath, 2019). Furthermore, we estimated the Bayesian Information Criterion (BIC), which similarly balances goodness of fit with model complexity but applies a stricter penalty for the number of parameters (Schwarz, 1978). A lower AIC and BIC indicate a better model, i.e., a better balance between goodness of fit and model complexity (Eq. 5). In practical terms, differences of more than 2 units in AIC or BIC are generally considered meaningful,

with larger differences (e.g., >10) providing strong evidence in favor of the model with the lower criterion value (Burnham & Anderson, 2002)

$$\text{AIC} = 2\text{k} - 2\ln(L)$$
$$\text{BIC} = \ln(\text{n})\text{k} - 2\ln(L) \tag{5}$$

where k is the number of parameters in the model, $L$ is the maximum of the log-likelihood function of the considered model, and n is the number of observations.

Once the optimal model was identified, the probability of occurrence of an event of a given magnitude and its return period (the inverse of the probability) can be estimated from the GEV distribution. The 90% confidence intervals (CI) of these quantities were obtained following a bootstrap approach, by repeating the fitting 1,000 times with random pairs of samples (TXx, covariate) drawn from the original sample with replacement.

### 2.2.2 Heat-related mortality attribution

To establish the relationship between temperature and mortality, we calculated the daily TX as a weighted averaged across the five meteorological stations, with weights based on the proportion of the population in each city. This weighted temperature was then analysed alongside the total number of daily deaths in the state of Rio de Janeiro over the period 2000-2019. While the primary focus is on the temperature-mortality relationship, we included time as a covariate in our analysis to account for important long-term trends and seasonal effects that could influence both temperature and mortality rates. Specifically, we fitted a generalized additive model with a negative binomial distribution, modeling time using a natural cubic spline with eight degrees of freedom per year (Ferreira et al., 2019). Moreover, we considered non-linear and time-lag effects by using distributed lag non-linear models (DLNM) (Gasparrini, 2011; Gasparrini et al., 2010). In this model, we selected a natural-spline with five degrees of freedom for the exposure-response function and a polynomial function with an intercept and four degrees of freedom for the lag-response function. This selection was made to enhance model flexibility, in accordance with the approach proposed by Ferreira et al. (2019). The model included lag estimates of up to 7 days (Tobías et al., 2023). A sensitivity analysis confirmed small effects of varying lags and degrees of freedom (Table S2). The resulting fit represents the exposure-lag-response associations, which capture the complex relationship between temperature exposure and mortality. This relationship is typically visualized as a U- or J-shaped curve (depending on the geographical location), whose minimum is designated as the minimum mortality temperature (MMT). To estimate the uncertainty, a bootstrapping procedure with 1,000 repetitions was employed (Tobías et al., 2017).

Using the coefficients estimated from the DLNM, we then calculated the daily attributable fraction (AF) of deaths due to excessive heat (Eq. 6), following the methodology proposed by Gasparrini & Leone (2014). The AF represents the proportion of deaths that can be attributed to heat exposure on a given day, considering both same-day and lagged effects. To compute the AF, we utilized the R code developed by Gasparrini & Leone (2014), implementing a backward perspective.

This approach links daily mortality to both current and past temperature exposures, capturing the cumulative and delayed effects of heat on mortality. Furthermore, the AF was estimated for hypothetical scenarios, considering the temperatures the event would have experienced in a pre-industrial climate and a future climate. These hypothetical temperatures were derived by accounting for the intensity changes of the event, as inferred from the non-stationary GEV model (Section 2.2.1). In this case, we assumed that the heat-mortality relationship is constant for all climate conditions, which allows a straightforward comparison of the potential effects of different levels of warming on mortality. This approach does not account for demographic changes (especially population aging) or adaptation (Lüthi et al., 2023).

$$AF_{backward} = 1 - e^{-\sum_{l=0}^{L} \beta_{x_{t-l},l}} \tag{6}$$

where $\beta_{x_{t-l},l}$ are the coefficients derived from the DLNM given an exposure x (i.e., temperature) at lag $l$, $t$ is the current day for which the AF is being calculated, $l$ ranges from 0 to L, where L is the maximum lag considered in the model, $x_{t-l}$ is the exposure on day $t$-$l$.

To estimate the confidence intervals of the daily AF, we accounted for two key sources of uncertainty, associated with the model coefficients and the temperature estimates under different climate scenarios. First, we generated 1,000 temperature perturbations based on the 90% confidence interval of temperature uncertainty. Then, for each temperature perturbation, we simulated 1,000 sets of model coefficients using the estimated variance-covariance matrix from the DLNM. For each combination of perturbed temperature and simulated coefficients, we recalculated the daily AF. This process resulted in a distribution of AF estimates that jointly captures the uncertainty from both temperature projections and model parameters.

## 3 Results

### 3.1 Contribution of Climate Change and ENSO to extreme temperatures

To put the 2023 hot days into context, the historical daily records of the TX series at the weather stations of Fig. 1 are analyzed (Table 1). In November 2023, all stations, except Cordeiro in the central region, registered new historical records in daily TX. In Cordeiro, TX was just one-tenth of a degree below the record of 2015. Furthermore, the city of Resende, in the west of the state, surpassed its previous historical TX, set in 1977, in two non-consecutive days. Remarkably, the southernmost station of Alto da Boa Vista beat its previous record, set in 1980, by 1 °C. The hottest day across the state of Rio de Janeiro was 18 November 2023, which broke all-time records in three out of the five stations, stressing the large spatial extent of the heatwave. The spatial average of TX across the five stations reached 39.4°C on that day, surpassing the temperature recorded on November 12, 2023, by 0.4°C and exceeding the estimated value for October 16, 2015, by 0.6°C. These findings highlight the exceptionally warm conditions experienced during November 2023. Given the proportion of missing data at some stations, we additionally verified that observations were available from all five stations on November

18, 2023. This confirmation strengthens our conclusion that the recorded maximum is based on actual observations and is not an artifact of gap-filling at any station.

**Table 1: Historical maximum temperature records in Rio de Janeiro. The two warmest days and their dates at the weather stations of the state of Rio de Janeiro based on the period 1971-2024, regardless of whether both occurred in the same year.**

| ID | Station | Warmest TX [°C] | Date of the warmest TX [YYYY-MM-DD] | Second warmest TX [°C] | Date of the second warmest TX [YYYY-MM-DD] |
|---|---|---|---|---|---|
| 83007 | Alto da Boa Vista | 39.6 | 2023-11-18 | 38.6 | 1980-12-05 |
| 83695 | Itaperuna | 42.8 | 2023-11-18 | 42.0 | 2012-10-31 |
| 83698 | Campos | 41.8 | 2023-11-12 | 41.6 | 2012-10-31 |
| 83718 | Cordeiro | 38.7 | 2015-10-16 | 38.6 | 2023-11-18 |
| 83738 | Resende | 39.6 | 2023-11-18 | 39.4 | 2023-09-24 |

To determine whether these outstanding TX values were isolated events or part of a broader warming trend, we examined the time series of TXx from 1971 to 2023 (Fig. 2). All stations, except Campos, exhibit a significant upward trend (at the 5% significance level) in the annual hottest day. Our findings indicate an increase of ~0.3°C per decade in TXx, with Itaperuna exhibiting the most pronounced trends within the state. However, it is important to note that a time interval of its series (1983–1989) was infilled using data from neighbouring stations. Although the infilled data lies in the mid years of the time series—preserving the observed endpoints and thus supporting the integrity of long-term trend estimation— uncertainty remains regarding the accurate representation of local extremes during this period. Consequently, while the strong trends observed at Itaperuna are robust in terms of sign, results for extreme values within the infilled interval should be interpreted with caution.

Besides the long-term trends, the time series depicted in Fig. 2 exhibit substantial inter-annual variability. The Spearman correlation between TXx and different ENSO indices (Table 2) was evaluated after filtering the trends of all series. This revealed strong links with ENSO at the two easternmost stations (Campos and Itaperuna), with El Niño favoring the increase in TXx. For the remaining stations, no significant correlation with ENSO was identified. Moreover, our findings are robust regardless of the ENSO index or temporal resolution considered, with no substantial changes observed in the results. It is worth mentioning that, by definition, the SOI index has the opposite sign to those based on SST; that is, an El Niño phase is associated with positive values in SST-based indices, while the SOI, which is based on atmospheric pressure, exhibits negative values during the El Niño phase.

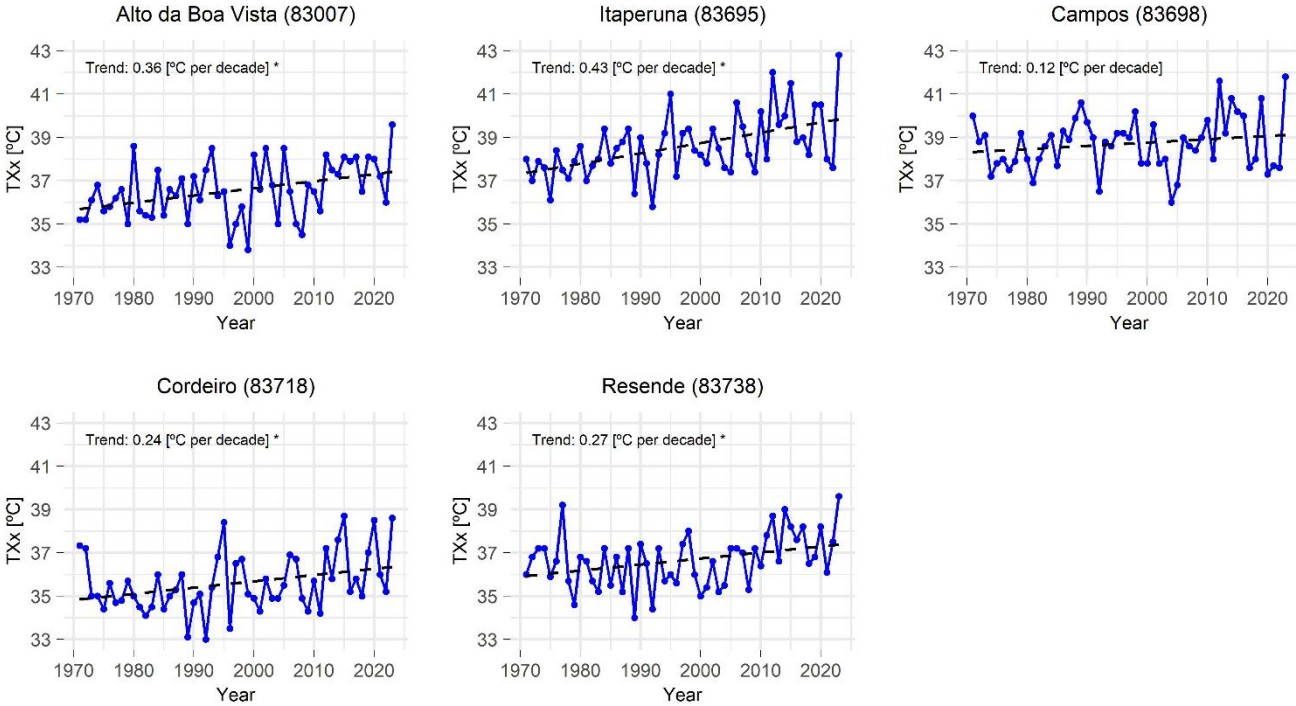

**Figure 2: Long-term trends in annual maximum daily temperatures (TXx) at weather stations in the State of Rio de Janeiro for the period 1971–2023. The figure shows linear trends (°C per decade) in the temperature of the hottest day recorded each year at each station. Stations with significant trends, as determined by Sen's slope and a Mann-Kendall test at the 5% significance level, are marked with an asterisk.**

Subsequently, the TXx data of each station were fitted to the GEV models described above. At all stations, the null hypothesis of the Kolmogorov-Smirnov test was not rejected for the stationary fit, suggesting that the observed data align well with the GEV distribution. However, this fit can be enhanced by incorporating additional covariates, based on the results of the LRT (Table S3), as well as the AIC and BIC (Table 3 and Table S4).

In Alto da Boa Vista and Cordeiro, the AIC suggests that the $M_{3\_multi}$ is only marginally better than the $M_{1\_gwi}$. Conversely, the BIC, which favors simpler models, identifies the GWI model as the optimal choice. Since the LRT does not indicate that one model significantly outperforms the other, and given the absence of a significant correlation between TXx and EN3.4 at these stations, we opted to use the simpler $M_{1\_gwi}$ in subsequent analyses. On the other hand, in the northernmost station of Itaperuna, characterized by a significant influence of the two covariates (GWI and EN3.4), the $M_{3\_multi}$ is the one with the lowest AIC and BIC values. In the easternmost station of Campos, the BIC suggests that the $M_{0\_s}$ is the optimal choice, while the AIC favors the $M_{3\_multi}$ as the better option. However, the latter was chosen based on the results of the LRT, which demonstrates that the model incorporating the two covariates significantly outperforms the stationary model, even in the absence of a significant trend in TXx. This indicates that incorporating GWI along with ENSO captures additional variability

that is not accounted for by ENSO alone, thereby enhancing the model's performance. Finally, in Resende, the $M_{1\_gwi}$ was chosen because both criteria coincide in indicating it as the best performing model. This is consistent with Figs. 2 and Table 2, which demonstrate a significant trend in TXx and no significant ENSO signals at this station.


**Table 2: Association between ENSO and extreme temperatures. Spearman correlation between the hottest day of the year (TXx) and different ENSO indices, after filtering the linear trends of both series. An asterisk indicates the significant correlations at 5%.**

| ID | Station | EN3.4 (monthly) | ONI (season) | SOI (monthly) | SOI (season) |
|---|---|---|---|---|---|
| 83007 | Alto da Boa Vista | 0.19 | 0.17 | -0.17 | -0.18 |
| 83695 | Itaperuna | 0.32 * | 0.33 * | -0.38 * | -0.38 * |
| 83698 | Campos | 0.31 * | 0.32 * | -0.26 | -0.22 |
| 83718 | Cordeiro | 0.19 | 0.21 | -0.24 | -0.22 |
| 83738 | Resende | 0.06 | 0.06 | -0.03 | -0.01 |

The parameters of the GEV distribution associated with the optimal models are shown in Table 4. For stations located further

east (Campos and Itaperuna), and no change in GWI, TXx increases by 0.2 to 0.4°C for each unit increase in EN3.4 index according to the $\beta_1$ coefficient. Regarding the relationship with the GWI, all stations except for Campos have a warming rate in TXx higher than that of the global mean temperature. In particular, Itaperuna stands out because it warms 2.5 times faster than the globe. Several factors may account for this warming rate, which highly exceeds the global average. Firstly, annual mean near-surface temperatures are increasing more rapidly over land than over the ocean, indicating a significant increase

in extreme land temperatures (Joshi et al., 2008; Sutton et al., 2007; Wallace & Joshi, 2018). Furthermore, the intensification of temperature variability in tropical land areas due to climate change exacerbates the warming of extreme temperatures (Olonscheck et al., 2021; Rehfeld et al., 2020).

The scale parameter, which reflects the dispersion of the extreme values in the fitted model, varies between 1.08 and 1.21°C, with maximum at the Alto da Boa Vista station (Table 4). The shape parameter of the GEV distribution is negative in our

analysis of temperature extremes, meaning that the probability of an event decreases rapidly as it approaches the upper boundary, and is zero above it (Wehner et al., 2018). The theoretical upper bounds of the extreme temperatures, as determined by the non-stationary GEV distributions, are presented in Fig. S3. It should be noted that these bounds are linearly dependent on the covariates under consideration for the shift of the GEV location parameter. In the pre-industrial climate, this upper limit for extreme temperatures is approximately 39°C, increasing to about 42°C in the present climate.

After fitting the best GEV model, the return periods of an event as intense as the 2023 TXx were estimated under different conditions of the covariates (GWI and ENSO). For the three stations where the $M_{1\_gwi}$ model is the optimal one, the return period is estimated under pre-industrial conditions (GWI = 0.00°C), the present climate (GWI of 2023, which has a value of 1.29°C) and a world two degrees warmer than the average temperature between 1850 and 1900 (GWI = 2.00°C). The latter

global warming level is in line with the goals set by the 2015 Paris Agreement (United Nations Framework Convention on
Climate Change, 2015).

**Table 3: The best GEV fit. Cells show the AIC and BIC for each GEV model. An asterisk indicates the best model by each criterion.**

|  | Criterion | $M_{0\_S}$ | $M_{1\_gwi}$ | $M_{2\_en3.4}$ | $M_{3\_multi}$ |
|---|---|---|---|---|---|
| Alto da Boa Vista (83007) | AIC | 182.91 | 175.72 | 182.70 | 175.69 * |
| | BIC | 188.82 | 183.60 * | 190.59 | 185.55 |
| Itaperuna (83695) | AIC | 188.25 | 177.74 | 186.19 | 173.40 * |
| | BIC | 194.16 | 185.62 | 194.07 | 183.25 * |
| Campos (83698) | AIC | 177.53 | 178.09 | 176.54 | 176.48 * |
| | BIC | 183.44 * | 185.97 | 184.42 | 186.33 |
| Cordeiro (83718) | AIC | 182.47 | 178.01 | 183.12 | 177.91 * |
| | BIC | 188.38 | 185.89 * | 191.01 | 187.76 |
| Resende (83738) | AIC | 176.24 | 169.89 * | 177.94 | 171.70 |
| | BIC | 182.15 | 177.77 * | 185.82 | 181.55 |

**Table 4: Estimated GEV parameters and standard errors for the best-fitting model based on maximum likelihood estimation. Asterisks denote parameters significantly different from zero at the 5% level (t-test).**

| ID | Station | Model | $\beta_0$ | $\beta_{1EN3.4}$ | $\beta_{2GWI}$ | Scale | Shape |
|---|---|---|---|---|---|---|---|
| 83007 | Alto da Boa Vista | $M_{1\_gwi}$ | 34.79 ± 0.48 * | | 1.81 ± 0.57 * | 1.21 ± 0.13 * | -0.33 ± 0.09 * |
| 83695 | Itaperuna | $M_{3\_multi}$ | 36.23 ± 0.49 * | 0.37 ± 0.14 * | 2.54 ± 0.64 * | 1.13 ± 0.12 * | -0.27 ± 0.10 * |
| 83698 | Campos | $M_{3\_multi}$ | 37.59 ± 0.47 * | 0.27 ± 0.14 | 0.88 ± 0.60 | 1.17 ± 0.12 * | -0.28 ± 0.09 * |
| 83718 | Cordeiro | $M_{1\_gwi}$ | 33.86 ± 0.49 * | | 1.62 ± 0.61 * | 1.11 ± 0.12 * | -0.15 ± 0.10 |
| 83738 | Resende | $M_{1\_gwi}$ | 34.92 ± 0.48 * | | 1.72 ± 0.58 * | 1.08 ± 0.11 * | -0.22 ± 0.08 * |

We start describing the results for the three stations having $M_{1\_gwi}$ as the best performing model. Figure 3 demonstrates that

the 2023 TXx event would have been virtually impossible under pre-industrial conditions in Alto da Boa Vista, with return

periods exceeding 10,000 years (Fig. 3a), and extremely rare in Resende, where return periods span from several hundred to thousands of years (Fig. 3c). In contrast, in Cordeiro, the 2023 TXx had a low probability of occurrence under a pre-industrial climate, with a likelihood of up to 1.5%, corresponding to a return period of 915.16 years (CI > 67.93 years) (Fig. 3b).

Nevertheless, the situation is substancially altered in the present climate, as the return periods for the 2023 TXx event

decrease considerably (Fig. 3). In both Alto da Boa Vista and Resende, what was once deemed nearly impossible under pre-industrial conditions is now expected to occur approximately once every 25 years (CI$_{\text{Alto da Boa Vista}}$ 11.06 — 4369.58 years, CI$_{\text{Resende}}$ > 10.22 years). This value is obtained by identifying the point where the horizontal line representing the observed 2023 event temperature intersects the present-day return period curve in Figure 3. Similarly, in Cordeiro, the return period has decreased to 19.30 years (CI 7.85 — 77.98 years).

Looking ahead to a future climate with a GWI of 2 °C, an event of these magnitudes (38.6 °C in Cordeiro and 39.6 °C in the other two stations) would occur every four to five years (CI$_{\text{Alto da Boa Vista}}$ 1.42 — 39.45 years, CI$_{\text{Cordeiro}}$ 1.54 — 27.67 years, CI$_{\text{Resende}}$ 1.34 — 27.69 years).

Moreover, events with these observed return periods (around 1-in-25 years) in the present climate are now about 2.2°C warmer compared to the pre-industrial climate (Fig. 3 and Table 5). Furthermore, in the future climate, such events are

projected to be about 1.2°C more intense than in 2023 at these stations (Fig. 3 and Table 5). Finally, the rate of change in the probability of an event like the one in 2023 between the current and future climates ranges from 3.9 times in Cordeiro to 7.9 times in Alto da Boa Vista, with Resende experiencing a 5.5-fold rise (Table S5). On average, this means that by the future, an event similar to that of November 2023 could be 5.8 times more likely to occur.

The remaining two stations, which are better described by M$_{\text{3\_multi}}$, exhibit a return period dependent on the two covariates

(Fig. 4). The slopes of the return periods vary with the station depending on the relative roles of GWI and EN3.4. At Campos station, where sensitivity to GWI is the lowest, the return period contour lines have steeper slopes compared to Itaperuna, indicating greater variability based on the ENSO phase at a fixed GWI.

In Itaperuna, the TXx recorded in 2023 would have been virtually impossible in a pre-industrial climate, regardless of the ENSO phase, since return periods of thousands of years are obtained (Fig. 4a). For present-day climate and La Niña

conditions (EN3.4 values lower than -0.5°C), the occurrence of this event would also have been highly unlikely (with return periods ranging from hundreds to thousands of years). However, under the observed El Niño conditions, the return period decreases to 31.55 years (CI > 7.59 years) in present-day climate, and the observed temperature was 3.3°C (CI 0.7 – 5.4°C) warmer than would be observed in a pre-industrial climate (Table 5 and Fig. S4). In the future climate, an event of the magnitude registered in 2023 would occur with a recurrence of 2.56 years for El Niño conditions (Fig. S4), and could even

be recorded under La Niña conditions with a return period of less than two decades (Fig. 4a). Furthermore, the probability of a TXx similar to that observed in 2023 is approximately 3.2% in the current climate. However, under El Niño conditions in a future climate, this probability rises to nearly 40%, making it 12 times more likely (Table S5). This positions Itaperuna as the location with the highest rate of increase.

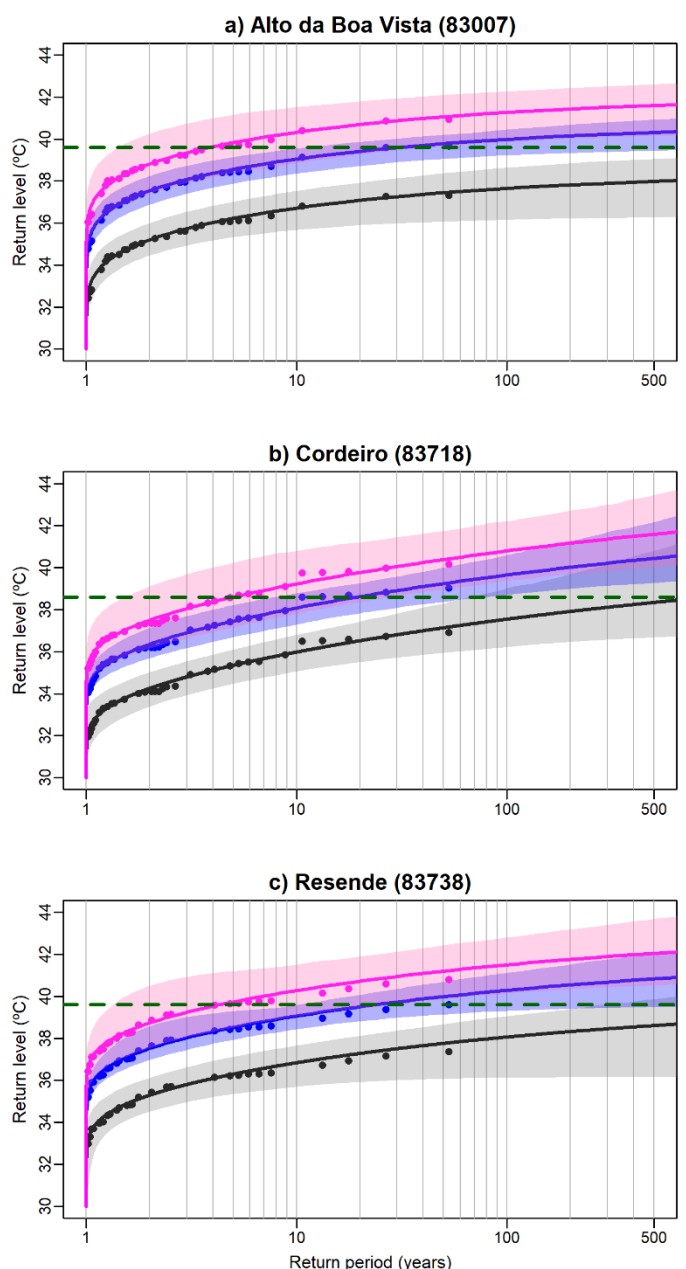

**Figure 3: Return period of extreme temperatures.** Frequency-magnitude curves of TXx under pre-industrial (black line, GWI = 0.00°C), present (blue line, GWI = 1.29°C) and future (magenta line, GWI = 2.00°C) climate according to a non-stationary GEV ($M_{1\_gwi}$) for Alto da Boa Vista (a), Cordeiro (b), and Resende (c) weather stations. The corresponding 90% confidence intervals are shown in shading. Observed extreme temperature values are plotted as points and shown three times—shifted to represent pre-industrial, present-day (2023), and future climates—by subtracting or adding the product of the global warming index (GWI) and the estimated GWI coefficient from the non-stationary GEV model's location parameter. The x-axis is displayed on a logarithmic scale. Horizontal dashed line denotes the magnitude of the observed event (in °C).

**Table 5: Change in the intensity of 2023-like events. Changes refer to events occurring with the same frequency as the 2023 TXx. The best GEV models are used for the estimation. For the multi-covariate model ($M_{3\_multi}$) and all climate conditions (GWI values), we consider an EN3.4 index equal to that observed in 2023 (EN3.4 = 2.02°C).**

| ID | Station | Model | Intensity (Preindustrial – Present) [°C] | Intensity (Future – Present) [°C] |
|---|---|---|---|---|
| 83007 | Alto da Boa Vista | $M_{1\_gwi}$ | -2.4 (CI -4.0 — -0.8) | 1.3 (CI -0.5 — 3.0) |
| 83695 | Itaperuna | $M_{3\_multi}$ | -3.3 (CI -5.4 — -0.7) | 1.8 (CI -0.9 — 4.4) |
| 83698 | Campos | $M_{3\_multi}$ | -1.1 (CI -3.7 — -0.7) | 0.7 (CI -1.8 — 3.4) |
| 83718 | Cordeiro | $M_{1\_gwi}$ | -2.1 (CI -3.8 — -0.4) | 1.2 (CI -0.9 — 3.4) |
| 83738 | Resende | $M_{1\_gwi}$ | -2.2 (CI -4.3 — -0.7) | 1.2 (CI -0.7 — 3.1) |

Differently, the 2023 TXx event in Campos could only have occurred in a pre-industrial climate under El Niño conditions, although it would have been an extremely unlikely event (return period of more than 1000 years, Fig. 4b). In the current climatic context, the return period strongly depends on the EN3.4 index, ranging from values in excess of a millennium under strong La Niña events to ~28 years for strong El Niño events (CI > 17.11 years). In terms of magnitude, the event would have expected to be 1.1°C (CI -1.2 – 2.7 °C) less intense if it would have occurred under strong La Niña conditions (Fig. S5). In the future, under stronger forcing, the range of variability in the return period associated with the ENSO phase is reduced and oscillates from approximately 50 to 5 years (Fig. 4b). Additionally, it is noted that Campos exhibits the smallest changes in the probability of the event between the future and the present climate under El Niño conditions (Table S5).

In summary, the ENSO contribution to the 2023 TXx is more relevant at Campos compared to the other stations, where ENSO plays a secondary or negligible role, especially weak at the westernmost stations. At these locations, the main driver of the increase in extreme temperature is the climate change signal, consistent with the findings of Kew et al. (2023).

## 3.2 Contribution of Climate Change to heat-related mortality

Figure 5 shows the daily evolution of the number of deaths in the state of Rio de Janeiro between July 2023 and June 2024. Total daily mortality fluctuated around 400 deaths per day throughout the year, and peaked to over 600 deaths on 18 November 2023, coinciding with the hottest day on record (see Section 3.1). Notably, other heat events of the 2023/24 season analyzed in earlier studies did not exhibit a comparable effect on mortality (Faranda & Alberti, 2024; Kew et al., 2023). These differences in mortality impact suggest disproportionate exposure to heat stress conditions before and during the 18 November 2023 event. In the seven days leading up to its peak, TX values exceeded or were close to the 95th percentile of the 2000–2019 period, indicating sustained stress on the human body (Fig. S6). In contrast, the September 2023 and March 2024 events lacked such prolonged extreme conditions, resulting in lower mortality.

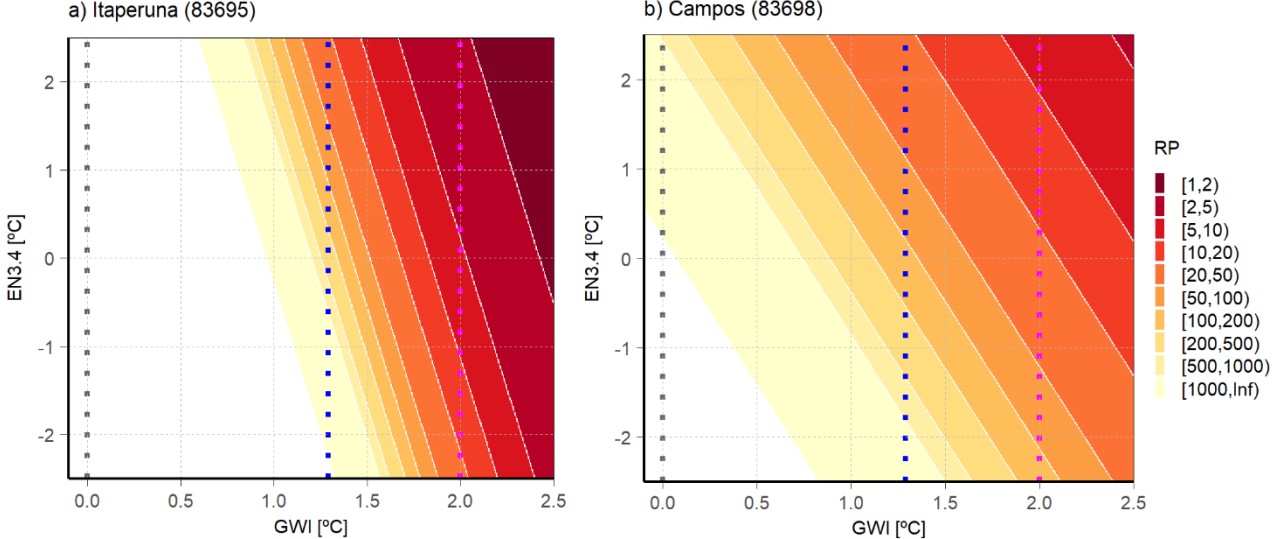

**Figure 4: Return period of extreme temperatures under two covariates. Panels denote the return period (in years) of the 2023 TXx as a function of the GWI (x-axis, in °C) and EN3.4 (y-axis, °C) indices obtained from the non-stationary GEV model ($M_{3\_multi}$). Dotted vertical lines indicate the pre-industrial (gray) and present-day (blue) climate conditions.**

For 18 November 2023, age-disaggregated data reveal that the elderly were disproportionately affected, with their daily mortality nearly doubling on that day compared to the annual average. In contrast, children under five years old were largely unaffected by the extreme heat (Fig. 5). In terms of gender, we found a significant increase in the proportion of female deaths on 18 November (51.42%) with respect to the rest of the year (49.14%) according to a proportions test (Infante Gil & Zarate de Lara 1984). As for the underlying causes, November 18 and 19 witnessed an increase in mortality associated with aggravation of circulatory diseases. Additionally, there was a notable increase in deaths linked to endocrine disorders, nervous system, and infections. Deaths categorized under "other causes" also escalated during this period (Fig. S7).

The cumulative relative risk as a function of TX for the state of Rio de Janeiro is shown in Fig. 6a. The curve is J-shaped reflecting the tropical climate of the region, and the risk increases with temperature, displaying a MMT at 27.6°C. This model is able to explain 68% of the variance of the data. On 18 November 2023, the state of Rio de Janeiro experienced an average TX of 39.4°C, which corresponds to a relative risk of mortality of 1.45 (CI: 1.37–1.52). Therefore, the risk of mortality was 45% higher than for the baseline temperature associated with minimum mortality risk (Fig. 6a). By examining the expected temperatures under different climatic conditions, we can better understand the amplified risks posed by global warming. In a pre-industrial climate, with event's temperatures approximately 2.3°C colder than recorded, the relative risk of mortality decreases to 1.30 (CI: 1.27–1.34). Conversely, in a future scenario with projected temperatures 1.3°C warmer than today, and no adaptation, the relative risk climbs to 1.53 (CI: 1.44–1.64), reflecting a marked increase in mortality risk. These results underscore the critical influence of climate change on the severity of heat-related health impacts.

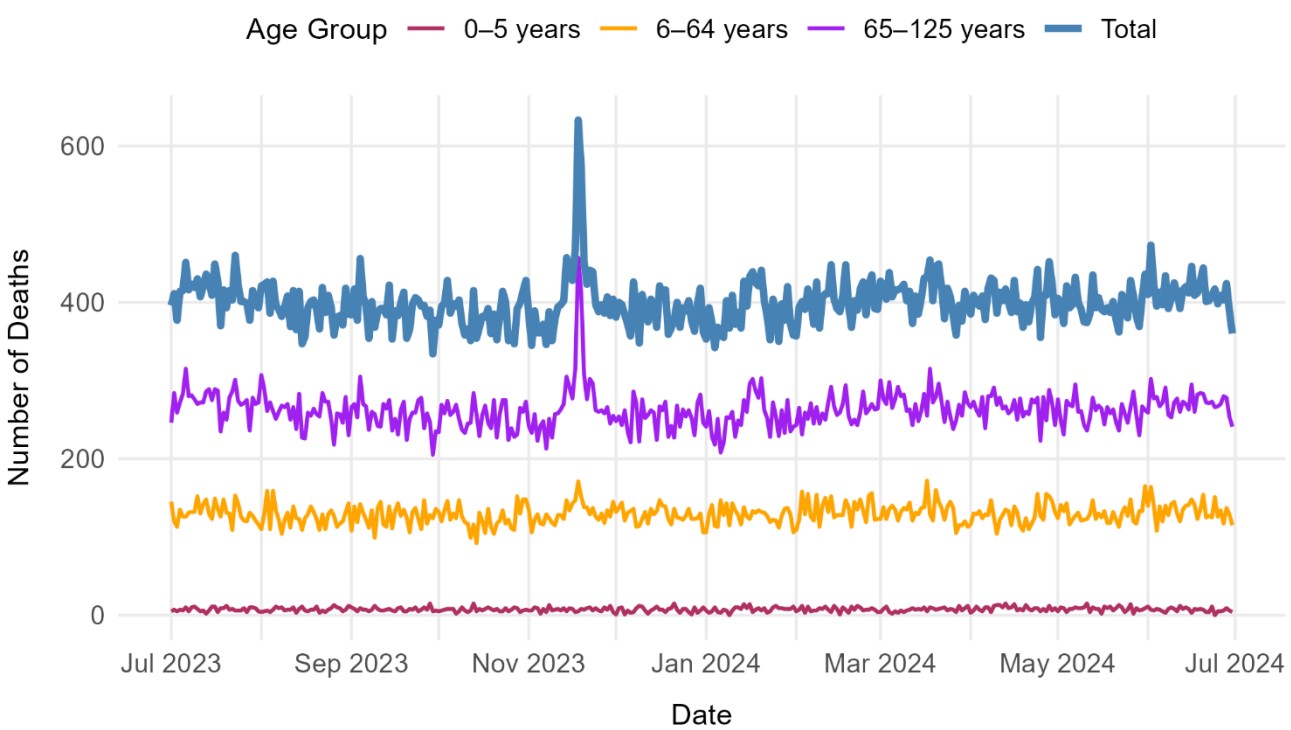

**Figure 5: Daily mortality for the state of Rio de Janeiro between July 2023 and June 2024, broken down by total and age ranges.**

Using the adjusted exposure-lag curves, we can estimate the daily proportion of deaths attributable to heat exposure (Fig. 6b). About 22.63% (CI: 19.70%–24.76%) of the 633 deaths recorded on 18 November 2023 are linked to heat exposure, equating to 143 deaths (CI: 125–157). Under pre-industrial conditions, the daily AF would have been 15.04% (CI: 10.91%–18.01%), about 8% lower than current levels, which highlights the impact of historical warming. A scenario 1.3°C warmer than 2023 is projected to result in a heat-related mortality rate of 26.76% (CI: 21.65%–30.92%), roughly 4% higher than today, emphasizing the additional health risks posed by future climate change if adaptation measures are not implemented.

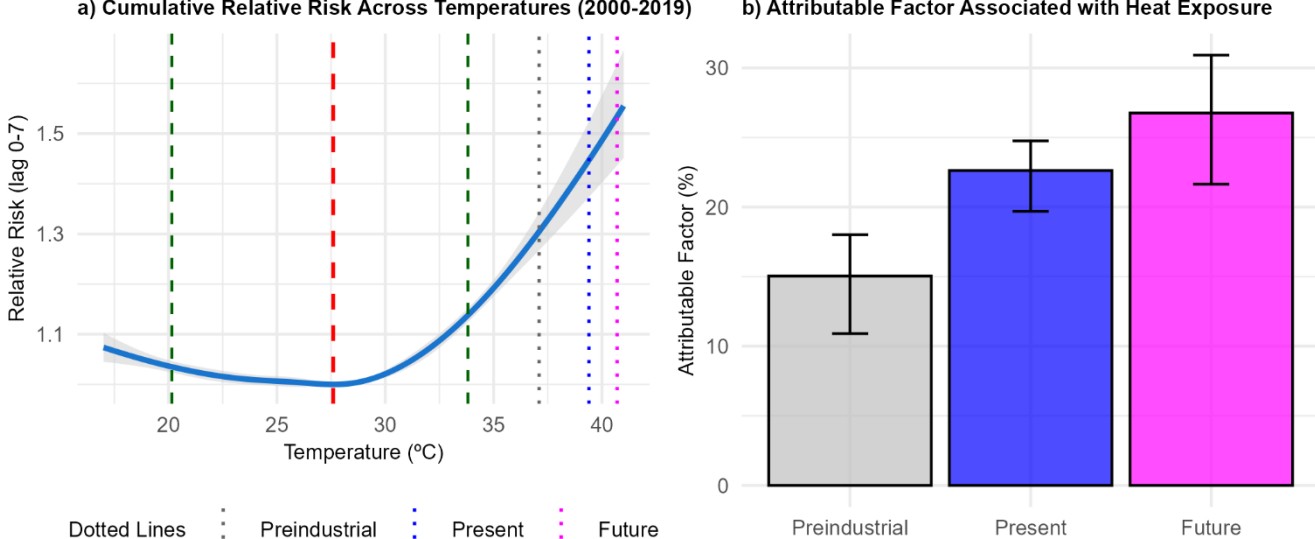

**Figure 6: Cumulated relative risks as a function of temperature for the State of Rio de Janeiro for the period 2000-2019 (a).**
**Shading denotes the 95% confidence interval inferred by a bootstrapping process (a). The Minimum Mortality Temperature -**
**MMT (red dashed vertical line), the 5th and 95th TX percentiles (green dashed vertical line), the TX observed on 18 November**
**2023 (blue dotted line), and estimated TX values under pre-industrial (grey dotted line) and future (magenta dotted line) climates**
**are also shown. Daily fraction of deaths attributable to heat exposure observed on 18 November 2023 for different levels of global**
**warming (b).**

**4 Discussion**

This study analyzed the role of global warming and ENSO in changing the probability of extreme temperatures in the State

of Rio de Janeiro. First, we evaluated whether these drivers effectively modulate TXx through linear trends and correlations.

We observed that the annual extreme warm temperatures have increased significantly over the last decades, except for the

Campos station. This finding is consistent with regional trends identified in previous studies for different periods and

datasets (Avila-Diaz et al., 2020; Regoto et al., 2021).

However, we observed regional differences in the magnitude of TXx trends, which may be due to distinct drivers. Byrne

(2021) demonstrated that the projected warming of temperature extremes is amplified over tropical lands, with peaks in the

interior of Brazil decreasing towards the coast. The author further asserted that the projected intensification of extreme

maximum temperatures on land is largely driven by the presence of drier soils. In this context, there is a marked increase in

the maximum number of consecutive dry days from the coast (~30 days) to the interior of the state (~50 days) (Luiz-Silva &

Oscar-Júnior, 2022). Additionally, recent studies report a significant trend in drought severity for cumulative water

imbalances on time scales of 12 months and longer (Tomasella et al., 2022). In addition, Cordeiro has registered the largest

precipitation reduction in the State of Rio de Janeiro over the period 1979-2009 (Sobral et al., 2019). This decline may have

contributed to the observed trends in TXx, as soil-atmosphere feedback mechanisms could have amplified these changes (Seneviratne et al., 2010). Another factor of intra-regional differences is the elevation of the stations (Table S1). Previous research has identified that the warming of annual mean temperature is considerably larger at higher (>500 m above sea level) than at lower altitude stations (Wang et al., 2016). Finally, we note that the urban heat island effect varies across stations, causing the global mean near-surface warming trend in the urban core to be 29% higher than the rural background trend (Liu et al., 2022, Table S1).

Regarding ENSO, considerable regional disparities are also observed in the correlation between the EN3.4 index and TXx. The EN3.4 index demonstrates a strong correlation with TXx in eastern regions (Itaperuna and Campos). However, this relationship weakens or becomes insignificant in western and central areas, indicating spatial heterogeneity in the influence of ENSO on extreme events within the state. These results corroborate previous studies that have performed similar analyses for elevated temperatures and drought and found a weak link between ENSO and climate variability in the state (de Oliveira-Júnior et al., 2018; Sobral et al., 2019; Wanderley et al., 2019). Furthermore, we tested different ENSO indices (ONI and SOI) and considered both monthly and seasonal scales, but found no substantial differences in the spatial distribution of correlations. This indicates that the observed spatial heterogeneity of ENSO's signal in TXx is robust regardless of the specific index or temporal resolution used.

Event attribution studies require examining extreme weather and climate-related events as they would occur in a world without human influence. Since observations of such a world are unavailable, all studies must rely on physical and statistical climate modeling, and thus these studies are dependent on the assumption that the model accurately simulates the specific weather event being studied (Otto, 2017). The approach employed in our work assumes a linear dependence of the location parameter on the covariates (i.e., a simple shift of the distribution), without additional changes in the shape and scale of the GEV distribution. This assumption is generally applicable to temperature data, but not for precipitation (Van Oldenborgh et al., 2018, 2022; Philip et al., 2020; Vautard et al., 2020). Applying this non-stationary model with GWI as a covariate allows us to obtain return period estimates for observed (present-day) events if they would have occurred in pre-industrial or future climates. Similarly, it is possible to infer changes in the magnitude of events that occur with a given frequency. In our study, we observed that the differences in TXx intensity between the pre-industrial and current periods between 0.5 and 4°C are consistent with the changes reported for the maximum temperatures of the September 2023 heatwave event in southeastern Brazil (Kew et al., 2023).

Furthermore, these non-stationary models could include additional covariates. While the relationship between TXx and South Atlantic SST was examined, no significant correlation at the 5% level was found at any station. As a result, it was not included as a covariate in the GEV distribution fit (Table S6). However, other local forcings—such as soil moisture content, local circulation patterns, topography, and proximity to the sea—as well as tropical variability processes like the Madden-Julian Oscillation (Alvarez et al., 2016) and the South American monsoon (Grimm, 2003), could be explored as potential covariates. It is crucial to acknowledge that the employed methodology does not account for uncertain changes in dynamic factors such as teleconnections, which may lead to an overestimation of attribution (Shepherd, 2016). Nonetheless, our

approach allows for a focus on the robust thermodynamic effects of climate change on the event (Beguería et al., 2023). Moreover, while an apparent scaling of the changes in teleconnections between ENSO and temperatures under different levels of warming was observed across much of Brazil, the state of Rio de Janeiro did not exhibit significant changes (McGregor et al., 2022). This would mean that no large changes in ENSO teleconnections are projected for the region, making our statistical approach more robust.

For the attribution of heat mortality, we used the well-established models, DLNM, which are flexible to fitting and capture non-linear and delayed effects to heat exposure (Ferreira et al., 2019; Gasparrini et al., 2010; Silveira et al., 2023; Tobías et al., 2023). While the 20-year mortality dataset used here may appear limited for climate attribution purposes, it aligns with and even exceeds the duration of many epidemiological studies examining temperature-mortality relationships, particularly in low- and middle-income countries where long-term health data are often sparse (MEASURE Evaluation, 2018). For example, recent multi-country analyses of heat-related mortality and its response to climate change have utilized observational periods ranging from 15 to 25 years, indicating that our analysed period is consistent with established methodologies in the field (Ballester et al. 2023, Lüthi et al. 2023). To make the analysis simple and interpretable, the model only establishes the relationship between TX and the total number of deaths in the state of Rio de Janeiro. We did not include relative humidity in the current analysis due to substantial missing data across stations (Table S7), which would have compromised the reliability of the results. Additionally, the estimated changes in mortality for different global warming levels (e.g., two degrees colder), should not be considered a predictive forecast. In future research, the possibility of including other variables, such as the cause of death and age group, will be explored. Similarly, the projections of the attributable mortality factor did not take into account population ageing, which has already been shown to increase the mortality burden (Chen et al., 2024), and adaptation.

**5 Conclusions**

In this paper, we have analyzed the contribution of El Niño-Southern Oscillation (ENSO) and climate change to the probability of daily maximum temperature (TX) extremes across five weather stations in the State of Rio de Janeiro by fitting a non-stationary GEV distribution. In addition, we have estimated changes in the magnitude and probability of occurrence of the record-breaking hot day of November 2023 for different (past and future) climate conditions and ENSO phases. The main findings can be summarized as follows:

- At all stations, the non-stationary GEV model significantly improved the fit to the annual TX maxima (TXx) compared to the stationary model. This improvement underscores the importance of including non-stationary elements that account for temporal changes in the characteristics of the data.
- The TXx series in the State of Rio de Janeiro exhibit substantial regional differences in their response to ENSO and climate change, probably influenced by the complex topography and proximity to the sea. The greatest response to

climate change is observed in Itaperuna, in the north of the state, while the relationship with ENSO maximizes in the east and progressively decreases towards the west.

- At the westernmost (Resende and Alto da Boa Vista) and central (Cordeiro) stations, the best non-stationary GEV model is the univariate one including global warming as the only covariate. At these stations, climate change has made 2023-like events ~2.2°C warmer than in the pre-industrial climate, when it would have been virtually impossible to record such a high TX.

- For stations in the eastern parts of the state (Itaperuna and Campos), the best fit is obtained with a multivariate non-stationary GEV. At these stations, both global warming and El Niño contributed to increasing the probability of occurrence of the observed 2023 TXx. Nevertheless, in none of the stations the ENSO effect overwhelms that of climate change. At these stations, climate change made 2023-like events up to 3.3°C warmer than in the pre-industrial climate.

- In a world that is two degrees warmer than the average temperature between 1850 and 1900, the return period of a TXx equal to 2023 is projected to be approximately one event every four years at all stations, except at Campos, where the return period is 9.29 years.

- The highest number of heat-related deaths in 2023 was recorded on the day when the absolute TX records were also documented. Climate change has made the daily heat-related attributable factor about 1.5 times higher than in a pre-industrial climate.

Therefore, climate change is likely the primary factor driving the increase in TXx in the current climate, with El Niño playing a secondary but measurable role, particularly at the two easternmost stations in the state. As global warming continues, the intensity of these events is expected to increase by more than 1°C, with the likely exception of the easternmost station (Campos), where a lower rate of warming has been observed in the historical period. Consequently, Rio de Janeiro will need to prepare for the associated impacts of the increased frequency of these extreme weather events (Geirinhas et al., 2021), such as disruptions to agriculture and water resources (Arreyndip, 2021; Luiz-Silva & Garcia, 2022), and increased risks to public health and infrastructure (Bitencourt et al., 2021; Prosdocimi & Klima, 2020). Proactive measures, including urban planning, public health initiatives and infrastructure resilience, will be essential to mitigate these challenges.

## Code availability

The code used in this work for fitting a non-stationary GEV to data from the state of Rio de Janeiro is available at https://github.com/SoleCollazo/Non-stationary-GEV NSGEV_rio_v2.1 (v2.1). https://doi.org/10.5281/zenodo.13913445

For the heat-related mortality analysis, we followed the code developed by Ferreira et al. (2019), available in the Supplementary Material.

## Data Availability Statement

Temperature station data for the State of Rio de Janeiro were provided by the Brazilian National Institute of Meteorology (INMET) upon request at the following Web site: https://bdmep.inmet.gov.br/#. Global mean temperature anomalies are available on the website of the Met Office of the United Kingdom: https://www.metoffice.gov.uk/hadobs/hadcrut5/.

ENSO is characterized from the El Niño 3.4 index available on the NOAA website: https://psl.noaa.gov/data/correlation/nina34.anom.data

Mortality data in the State of Rio de Janeiro are publicly available, provided by the Secretaria de Estado de Saude of Rio de Janeiro on its website (http://sistemas.saude.rj.gov.br/tabnetbd/dhx.exe?sim/tf_sim_do_geral.def).

## Author Contributions

S.C. performed data curation, formal analysis, visualization, funding acquisition, investigation, software development, and wrote the original draft. D.B. contributed to conceptualization, formal analysis, and review & editing. R.G-H. was involved in conceptualization, supervision, and review & editing. S.B. contributed to methodology, software development, and review & editing. All authors reviewed the manuscript.

## Competing interests

The authors declare that they have no conflict of interest.

## Acknowledgments

This work was supported by the SAFETE project, which has received funding from the European Union's Horizon 2020 research and innovation program under the Marie Skłodowska-Curie grant agreement No 847635 (UNA4CAREER). The authors are grateful to the Brazilian National Institute of Meteorology (INMET) for providing the data. This research work was supported by the Ministry for the Ecological Transition and the Demographic Challenge (MITECO) and the European Commission NextGenerationEU (Regulation EU 2020/2094), through CSIC's Interdisciplinary Thematic Platform Clima (PTI-Clima).

**Financial support**

This work was supported by the SAFETE project, which has received funding from the European Union's Horizon 2020 research and innovation program under the Marie Skłodowska-Curie grant agreement No 847635 (UNA4CAREER); and Ministry for the Ecological Transition and the Demographic Challenge (MITECO) and the European Commission NextGenerationEU (Regulation EU 2020/2094), through CSIC's Interdisciplinary Thematic Platform Clima (PTI-Clima).

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
