# Peer review of "Extreme heat and mortality in the State of Rio de Janeiro in November 2023: attribution to climate change and ENSO"

_EGUsphere, 2025_

## Referee Comment (RC1)

Why did authors use the 15% threshold for showing missing data points "less than 15% of missing data in the period 1971-2024 are identified with a black dot and the name?" The second warmest day in 83738 Resende appear to be around 1976 (Figure 2) however it does not match with Table 1 with its reporting in 2023. Similarly, peaks in temperature appears much before as shown in Figure 2 than its reporting in Table 1 (83718 Cordeiro). The ENSO is itself impacted by climate change, then how do authors decouple the separate impact of climate change and ENSO. In addition, without a scientific evidence it is vague to say or quantify how they both impact extreme heat. Mortality is heavily driven by extreme heat in combination with higher humidity which is not at all explored. The figure qualities are also inadequate and not suitable for scientific standard publications.

---

## Author Comment (AC1)

Reviewer 1:

Thank you for taking the time to review our work and for your valuable comments, which have helped us enhance the quality of our study.

1) Why did authors use the 15% threshold for showing missing data points "less than 15% of missing data in the period 1971-2024 are identified with a black dot and the name?"

We used a 15% threshold for missing data to balance data quality and station coverage across the state. We clarified this in the text and added a supplementary figure S1 to show the temporal evolution of missing data at Itaperuna (the station with the highest percentage of missing data in the period 1971-2024), as well as explanations about how these missing data might affect the results.

Lines 287-293: "Our findings indicate an increase of ~0.3°C per decade in TXx, with Itaperuna exhibiting the most pronounced trends within the state. However, it is important to note that a time interval of its series (1983–1989) was infilled using data from neighbouring stations. Although the infilled data lies in the mid years of the time series—preserving the observed endpoints and thus supporting the integrity of long-term trend estimation—uncertainty remains regarding the accurate representation of local extremes during this period. Consequently, while the strong trends observed at Itaperuna are robust in terms of sign, results for extreme values within the infilled interval should be interpreted with caution."

2) The second warmest day in 83738 Resende appear to be around 1976 (Figure 2) however it does not match with Table 1 with its reporting in 2023. Similarly, peaks in temperature appears much before as shown in Figure 2 than its reporting in Table 1 (83718 Cordeiro).

The apparent mismatch between Table 1 and Figure 2 arises because they present different types of information. Figure 2 displays only the *annual* maximum temperature (TXx) for each year at each station, so it highlights the single hottest day per year. In contrast, Table 1 lists the two highest daily temperature records over the entire period (1971–2024), regardless of whether they occurred in the same year or different years. This means that if both the first and second highest temperature records at a station happened in the same year, only the highest will be shown in Figure 2, but both will be reported in Table 1. Therefore, it is possible —and expected— for Table 1 to show record dates that do not appear in Figure 2.

3) The ENSO is itself impacted by climate change, then how do authors decouple the separate impact of climate change and ENSO. In addition, without a scientific evidence it is vague to say or quantify how they both impact extreme heat.

Thank you for the comment. We acknowledge that climate change can influence ENSO behavior, and the interactions between these processes remain an area of active research, being subject to considerable uncertainty. Therefore, disentangling the effects of climate change and ENSO on extreme heat events is inherently challenging. We recognize the complexity of these interactions and have taken methodological steps to minimize confounding effects as much as possible. In particular, we addressed this issue statistically by applying a LOESS filter to the global mean temperature series to remove

interannual variability, thereby isolating the long-term climate trend. Similarly, we removed trends from the ENSO indices to focus on their interannual fluctuations. Furthermore, we found no significant correlation between the detrended global mean temperature and ENSO indices, indicating that, within our framework, their impacts on extreme heat events can be considered largely independent.

To assess the separate influences of climate change and ENSO on extreme heat, we performed a structured statistical analysis. First, we identified the meteorological stations in Rio de Janeiro state that exhibited significant trends in annual maximum temperature (TXx). We then calculated correlations between TXx and various ENSO indices (e.g., Niño 3.4) to evaluate the potential influence of ENSO phases on temperature extremes. Following this, we applied the Generalized Extreme Value (GEV) distribution to model TXx, testing several versions of the model: a stationary GEV (no covariates), a time-dependent model, and fully non-stationary models incorporating both a long-term trend (as a proxy for climate change) and ENSO indices as covariates. Model comparison using information criteria (AIC) and goodness-of-fit diagnostics showed that the non-stationary GEV models provided better performance, indicating that these two independent covariates (climate change and ENSO) contribute significantly to the behavior of temperature extremes. Therefore, the results presented in the original manuscript offer quantitative evidence of the individual influences of these two factors, which together lead to an improved representation of extreme heat in the region.

4) Mortality is heavily driven by extreme heat in combination with higher humidity which is not at all explored.

We fully agree with the reviewer that humidity, in combination with extreme heat, plays a crucial role in influencing mortality rates. Unfortunately, the inclusion of relative humidity in our analysis was not feasible due to substantial data gaps in the INMET records for the studied weather stations. As shown in the table below, the percentage of missing data for both daily mean and minimum relative humidity is unacceptably high at most stations, particularly at Alto da Boa Vista, where over 87% of the data are missing. Given these limitations, any analysis incorporating humidity would be hampered by the lack of data, thus preventing reliable conclusions. Note that none of the stations satisfy the aforementioned 15% threshold for missing data. We have added a comment about this limitation in the Discussion section and a Supplementary Table.

Lines 530-532: "We did not include relative humidity in the current analysis due to substantial missing data across stations (Table S7), which would have compromised the reliability of the results."

| Station | Variable | Missing Data [%] |
|---|---|---|
| Alto da Boa Vista | Daily mean relative humidity | 88.20 |
| | Daily minimum relative humidity | 87.11 |
| Itaperuna | Daily mean relative humidity | 22.72 |

| | | |
|---|---|---|
| | Daily minimum relative humidity | 22.02 |
| Campos | Daily mean relative humidity | 30.00 |
| | Daily minimum relative humidity | 25.33 |
| Cordeiro | Daily mean relative humidity | 46.73 |
| | Daily minimum relative humidity | 31.86 |
| Resende | Daily mean relative humidity | 33.02 |
| | Daily minimum relative humidity | 27.19 |

Table S7: Percentage of missing data in relative humidity variables for the five stations studied in the INMET dataset.

5) The figure qualities are also inadequate and not suitable for scientific standard publications.

The conversion of the file to pdf decreased the quality of the figures. We will try to improve it in this new version and/or include the figures in individual files with high resolution.

---

## Author Comment (AC2)

Reviewer 2:

Thank you for taking the time to review our work and for your valuable comments, which have helped us enhance the quality of our study.

**Major comments**

1) According to what you say on p.5. l.132. you consider for each year only the monthly anomaly of this index for the month determining the TXx value of each year. By considering the index monthly and not seasonally, there are intra-seasonal variations that can influence the results and that do not necessarily account for the phenomenon you are trying to describe, which manifests itself on predominantly seasonal scales.

To address the reviewer's concern, we conducted sensitivity tests using **3-month centered averages of ENSO indices** and compared the results to our original monthly-index approach. These tests did not reveal substantial differences. In this revised version, Table 2 shows the correlations between TXx and the different ENSO indices. Using the seasonal index, we came to the same conclusion: the stations in the eastern part of the state are the most influenced by ENSO. Likewise, the selection of the GEV model (stationary, univariate, or multivariate) with the best fit to the data does not change substantially when modifying the ENSO index used (see new Table S4).

2) Another observation in the same line is that, considering the month in which TXx is obtained and the seasonal cycle of SSTs in the tropical Pacific, the month of the year in which TXx is given can have an important influence.

Thank you for the observation. We would like to clarify that, due to the tropical location of the state of Rio de Janeiro, TXx can occur over a broad range of months, generally from September to May. This is consistent with the region's climate, which features warm temperatures throughout most of the year and does not have a sharply defined hot season limited to only a few months. As a result, the timing of TXx events is not restricted to a narrow seasonal window, and indeed ENSO can vary greatly within this extended warm season. That is why we initially focused only on the ENSO index value of the month with the occurrence of TXx. Following the reviewer's suggestion, we have repeated the analyses using the seasonal ENSO index. The results do not vary significantly when considering the SSTs of the three months around the TXx event (Table 2 and S4).

3) What is the period over which the anomalies are calculated, and can there be a bias in recent years due to the observed trend?

The anomalies for the ENSO indices in our study are calculated using the 1991–2020 base period, following the current operational standards followed by institutional data providers such as NOAA. We now clarify this choice in the revised manuscript.

4) In the link provided I only see data from 1982. Was this the only period used for the adjustment?

We used ENSO data for the full period 1971–2024, consistent with the temporal coverage of the maximum temperature data in our study. The full dataset, including values from 1971, is now available at the updated NOAA PSL link:

https://psl.noaa.gov/data/correlation/nina34.anom.data. We have changed the link in the main text.

5) Can the results be improved by considering an ENSO index that takes into account the atmospheric part of ENSO (e.g. SOI or MEI)?

To explore this, we considered the Southern Oscillation Index (SOI), which reflects the atmospheric component of ENSO. The results did not show a considerable improvement over the SST-based indices. Specifically, in one of the two stations with an ENSO signal (Campos station) the relationship with ENSO weakens when using the SOI compared to an SST-based index, while in Itaperuna, the opposite occurs (Table 2). These contrasting responses suggest that incorporating the atmospheric component does not systematically enhance the analysis across the study area (Table S4).

6) In summary, I think it would be important for you to assess whether you are considering the effect of ENSO in the most appropriate way and that the concerns that I raised above could not significantly change the results. Some discussion of the limitations of methodological choices should also be included in the text.

We thank the reviewer for this thoughtful summary and agree on the importance of carefully assessing the influence of ENSO and the methodological choices involved. In response, we conducted additional sensitivity analyses using seasonally averaged indices and alternative ENSO metrics (e.g., SOI), as detailed in our responses above. These tests did not reveal substantial changes in the main results, indicating that our conclusions are robust to these methodological variations. We have added a comment about this in the Discussion section:

Lines 494-497: "Furthermore, we tested different ENSO indices (ONI and SOI) and considered both monthly and seasonal scales, but found no substantial differences in the spatial distribution of correlations. This indicates that the observed spatial heterogeneity of ENSO's signal in TXx is robust regardless of the specific index or temporal resolution used."

7) Table S1 shows that there are large differences between the populations of the different cities for which each station is representative. In this sense, Would it be logical to calculate the mean daily TX for the five weather stations in order to establish a correlation between temperature and mortality? Would it be more appropriate to consider a method of analysing the Tx that incorporates a weighting based on these population differences?

Thank you for this valuable suggestion. Following the reviewer's recommendation, we recalculated the daily maximum temperature (TX) as a weighted average across the five meteorological stations, using the proportion of the population in each city as weights. The results based on this population-weighted TX were very similar to those obtained using the simple TX mean, with no substantial changes in the estimated associations between temperature and mortality. We have updated the methods section using this new approach.

Lines 228-231: "To establish the relationship between temperature and mortality, we calculated the daily TX as a weighted averaged across the five meteorological stations,

with weights based on the proportion of the population in each city. This weighted temperature was then analysed alongside the total number of daily deaths in the state of Rio de Janeiro over the period 2000-2019."

8) I recognise the effort to obtain results related to the impact of anthropogenic warming on health in such an extreme event and the limitations of data availability, but I wonder how robust the results are if only a 20-year period is considered in order to hypothesise what would happen at other levels of global warming.

While the 20-year mortality dataset used here may appear limited for climate attribution purposes, it aligns with and even exceeds the duration of many epidemiological studies examining temperature-mortality relationships, particularly in low- and middle-income countries where long-term health data are often sparse. Recent multi-country analyses of heat-related mortality and its response to climate change have utilized observational periods ranging from 15 to 25 years, indicating that our analyzed period is consistent with established methodologies in the field (Ballester et al. 2023, Lüthi et al. 2023). As is common in health research, prolonged and comprehensive data availability is often limited due to technical, organizational, and methodological barriers, which can impact the scope and duration of analyses (Bernardi et al. 2023).

Like in all attribution studies, the construction of counterfactuals was primarily *hypothetical*—aimed at exploring how mortality might have differed if the same event had occurred under colder or warmer climate conditions. This approach does not rely on extrapolating long-term trends but instead examines the *relative* change in mortality under altered magnitudes of the event. We have clarified these points in the revised manuscript:

Lines 522-536: "For the attribution of heat mortality, we used the well-established models, DLNM, which are flexible to fitting and capture non-linear and delayed effects to heat exposure (Ferreira et al., 2019; Gasparrini et al., 2010; Silveira et al., 2023; Tobías et al., 2023). While the 20-year mortality dataset used here may appear limited for climate attribution purposes, it aligns with and even exceeds the duration of many epidemiological studies examining temperature-mortality relationships, particularly in low- and middle-income countries where long-term health data are often sparse (MEASURE Evaluation, 2018). For example, recent multi-country analyses of heat-related mortality and its response to climate change have utilized observational periods ranging from 15 to 25 years, indicating that our analyzed period is consistent with established methodologies in the field (Ballester et al. 2023, Lüthi et al. 2023). To make the analysis simple and interpretable, the model only establishes the relationship between TX and the total number of deaths in the state of Rio de Janeiro. We did not include relative humidity in the current analysis due to substantial missing data across stations (Table S7), which would have compromised the reliability of the results. Additionally, the estimated changes in mortality for different global warming levels (e.g., two degrees colder), should not be considered a predictive forecast. In future research, the possibility of including other variables, such as the cause of death and age group, will be explored. Similarly, the projections of the attributable mortality factor did not take into account population ageing, which has already been shown to increase the mortality burden (Chen et al., 2024), and adaptation."

Bernardi, F. A., Alves, D., Crepaldi, N., Yamada, D. B., Lima, V. C., & Rijo, R. (2023). Data Quality in Health Research: Integrative Literature Review. Journal of medical Internet research, 25, e41446. https://doi.org/10.2196/41446

**Minor comments**

- Title: As the study finally focuses on the November 2023 event, it seems more correct to me to remove the reference to 2024 from the title and other relevant parts.

  We have removed the reference to 2024, as suggested.

- p.1 l. 17-18 change to "as a function of global warming and/or El Niño-Southern Oscillation (ENSO)"

  Done

- p.1 l.20-24. I think the words you use in these lines like 'heatwaves' or '2023-like daytime temperatures' are not quite accurate to what you actually assessed in the study. So I recommend you rewrite it.

  We changed it. Lines 20-23: "Events as likely as the 2023 record were estimated about 2°C colder in pre-industrial times. Under a 2°C global warming scenario, the probability of experiencing maximum temperatures equal to the 2023 increases by at least a factor of three."

- p.1 l.25. I suggest also adding 'mitigation measures'

  Done

- p.1 l.30 Please clarify which climatology.

  We clarified the climatological period.

- p.1 l.31 Does November not belong to spring?

  Yes, November is part of spring. We rephrased the sentence to avoid misunderstanding.

  Lines 30-33: "This period recorded the warmest spring in at least 63 years for the region, with TX locally exceeding 43°C, which was 5-8°C higher than the 1991-2020 climatology (Kew et al., 2023; Perkins-Kirkpatrick et al., 2024). The intense heat persisted throughout the season and peaked in November, when TX anomalies reached +9°C in some areas of southern Brazil."

- p2. l35-37. The impact of El Niño events on your region of interest is also (and probably more) related to Rossby wave trains.

  We have added an explanation about ENSO teleconnections through Rossby waves in the revised manuscript.

Lines 38-41: "In addition to modifications in the Walker circulation, ENSO-related impacts over South America are also modulated by tropical-extratropical teleconnections. This mechanism involves stationary Rossby wave trains, initiated by anomalous convection over the tropical Pacific, which propagate into the mid-latitudes, generating alternating centres of high and low atmospheric pressure (Cai et al., 2020)."

- p.2 l.43. Please check what is the recommended way to cite a specific chapter of the IPCC AR6.

  In this new version we have quoted from the IPCC Summary for Policymakers.

- p.4 l.99. Table S1 shows that Itaperuna has 15.00% missing data. Is it really below the selection criteria if we have more digits? Moreover, the missing data for this station is relatively high, both for the whole period and for the period of interest, to what extent could this feature affect the results, especially considering that you are working with extremes? I think it would be desirable to have a discussion or a note of caution regarding this.

  Table S1 reports 15.00% missing data for Itaperuna, which is close to our selection threshold. When considering more decimals, the actual value remains below the established cutoff (14.996%), so the station was included according to our criteria.

  In the revised version, we have added a new figure in the supplementary material showing the percentage of missing data by year for the Itaperuna station (Fig. S1). The period 1983-1989 concentrates the highest amount of missing data. We have discussed in the text how this affects the estimation of the trend, further confirming that all stations have valid records for the historical record day of November 18, 2023.

  Lines 287-293: "Our findings indicate an increase of ~0.3°C per decade in TXx, with Itaperuna exhibiting the most pronounced trends within the state. However, it is important to note that a time interval of its time series (1983–1989) was infilled using data from neighbouring stations. Although the infilled data lies in the mid years of the time series—preserving the observed endpoints and thus supporting the integrity of long-term trend estimation—uncertainty remains regarding the accurate representation of local extremes during this period. Consequently, while the strong trends observed at Itaperuna are robust in terms of sign, results for extreme values within the infilled interval should be interpreted with caution."

- p.4 l.117. Please explain further how you apply the block maxima approach.

  We have added an explanation on how the block maxima approach was applied.

  Lines 126-130: "Finally, we identified the extreme values of the completed TX series by using the block maxima approach. This method involves dividing the time series into non-overlapping blocks, and selecting the highest daily maximum temperature within each block. As the region under study is located within a tropical climate zone, we used annual blocks, therefore taking the TX value of the hottest day of the year (TXx)."

- p.5. l-123 Please specify the parameters used for the LOESS smoothing.

  We have specified the parameters for the LOESS smoothing, as follows:

Lines 135-137: "This LOESS model applies a smoothing span of 0.75, which determines the proportion of data used in each localized fit. It employs a second-degree polynomial for local regression, and assumes normally distributed errors."

- p.6 l.174. Please include the references of those studies that use this approach for South America.

  We have added a reference to Pereira et al. (2023), who applied non-stationary GEV models to extreme temperature analysis in Campinas, Brazil, using time as the sole covariate. To our knowledge, our approach, modeling both single and combined effects of GWI and EN3.4 as covariates, remains little explored in South American attribution studies. This has been clarified in the revised manuscript.

  Lines 189-192: "Considering single and combined influences of two covariates is an approach little explored in South American attribution studies. For example, Pereira et al. (2023) applied non-stationary GEV models to extreme temperature analysis in Campinas, Brazil, using time as the sole covariate to model changes in the location parameters."

  **Reference:**
  Pereira, L. B., Martins, L. L., Rodrigues, I. C. A., Sobierajski, G. R., & Blain, G. C. (2023). Changes in Extreme Air Temperature in One of South America's Longest Meteorological Records: Campinas, Brazil (1890–2022). *Bragantia*, 82, e20230128. https://doi.org/10.1590/1678-4499.20230128

- p.7 l.195-199. Can you give some indication of how these numerical values are interpreted?

  We have added some explanations about this.

  Lines 213-216: "A lower AIC and BIC indicate a better model, i.e., a better balance between goodness of fit and model complexity (Eq. 5). In practical terms, differences of more than 2 units in AIC or BIC are generally considered meaningful, with larger differences (e.g., >10) providing strong evidence in favor of the model with the lower criterion value (Burnham & Anderson, 2002)"

- p.14 l.333. Where exactly did you get the '25 years' value, is it the mean around the 3 stations? In order to visualise what you said more easily, I think it might help to clearly indicate (with a different colour or higher line width) the horizontal line corresponding to this value in Figure 4.

  The '25 years' value is determined by the intersection of the horizontal line indicating the event temperature and the present-day return period curve in Figure 3. This intersection gives the estimated return period for the observed extreme event under current climate conditions. We have clarified this in the main text.

  Lines 365-368: "In both Alto da Boa Vista and Resende, what was once deemed nearly impossible under pre-industrial conditions is now expected to occur approximately once every 25 years ($CI_{\text{Alto da Boa Vista}}$ 11.06 — 4369.58 years, $CI_{\text{Resende}}$ > 10.22 years). This value is obtained by identifying the point where the horizontal line representing the observed 2023 event temperature intersects the present-day return period curve in Figure 3."

- p.14 l.355. I think that '(Fig. 5c)' is not the correct reference to what you are saying.

  The reviewer is right, the figure indicated was not the correct one. We have corrected and revised the references to Tables and Figures.

- p.17 l.396 "In terms of gender, the usual mortality pattern was observed". Which is the usual mortality pattern?

  In the revised manuscript, we have rephrased the sentence and conducted a statistical test to determine if the mortality increase (by gender category) is meaningful. The results revealed a statistically significant increase in the proportion of female deaths on November 18 compared to the rest of the year. The revised text is shown below:

  Lines 436-438: "In terms of gender, we found a significant increase in the proportion of female deaths on November 18 (51.42%) with respect to the rest of the year (49.14%) according to a proportions test (Infante Gil & Zarate de Lara 1984)."

- p.18 l.414-420. How do you calculate the CI for daily AF? Is it based on the uncertainty of the methodology you used or the uncertainty associated with the estimation of Tx for future conditions? I think it is important to take both uncertainties into account.

  Thank you for this valuable comment. In the previous version of the manuscript, the confidence intervals for the daily attributable fraction (AF) were based solely on the uncertainty in the temperature projections, as we observed that this source of uncertainty was greater than that associated with the methodology itself. However, in the revised version, we now account for both sources of uncertainty: (1) the variability in temperature estimates and (2) the uncertainty associated with the statistical model used to estimate AF.

  To do so, we implemented a two-step simulation approach. First, we simulated 1000 sets of model coefficients using the estimated variance-covariance matrix of the fitted model. Then, for each simulated model, we introduced temperature perturbations sampled from the 90% confidence interval of temperature uncertainty, as estimated in the previous section using the return period of the 2023 event. For each combination, we calculated the daily AF, resulting in a distribution of AF estimates from which we derived updated confidence intervals (e.g., 5th and 95th percentiles).

  This method allows us to incorporate both sources of uncertainty more robustly in our estimation of the daily AF. We have incorporated this explanation in the methodology section:

  Lines 261-266: "To estimate the confidence intervals of the daily AF, we accounted for two key sources of uncertainty, associated with the model coefficients and the temperature estimates under different climate scenarios. First, we generated 1,000 temperature perturbations based on the 90% confidence interval of temperature uncertainty. Then, for each temperature perturbation, we simulated 1,000 sets of model coefficients using the estimated variance-covariance matrix from the DLNM. For each combination of perturbed temperature and simulated coefficients, we recalculated the daily AF. This process resulted in a distribution of AF estimates that jointly captures the uncertainty from both temperature projections and model parameters."

- p.19 l.446. '…causing the global mean near-surface warming trend in the urban core…"

Done

- p.26 l.666-668 Kew et al (2023) reference is incomplete.

We have completed the reference

- Faranda & Alberti (2024) reference is missing

We added the reference.

- Please unify the spelling of 'heat wave'/'heatwave'

Done

- Figure 1: Add a source for the elevation.

We added the source for the elevation in the caption "Elevation data were accessed via the Amazon Web Services Open Data Terrain Tiles using the elevatr R package (Hollister et al., 2023)"

- Figure 2: In the caption, be more specific about what you are plotting.

We have changed the caption of Figure 2, as follows:

"Figure 2: Long-term trends in annual maximum daily temperatures (TXx) at weather stations in the State of Rio de Janeiro for the period 1971–2023. The figure shows linear trends (°C per decade) in the temperature of the hottest day recorded each year at each station. Stations with significant trends, as determined by Sen's slope and a Mann-Kendall test at the 5% significance level, are marked with an asterisk."

- As you have only five points, I would suggest replacing Fig. 3, Fig. 5, Fig. S2 and Fig. S4 with tables. This will make it clear which is the exact value of the corresponding variable and also you cloud include there other information such as the CI. Should you consider it worthwhile, you may wish to include a concise description to provide an indication of the location (e.g. 'West', 'South', etc.).

We have replaced the figures with tables and included confidence intervals.

- I suggest that the information in Figure S2 be included as a table in the main text, as it contains key information for the reproduction of the study.

Done

- Figure 4 (and other similars): Could you better explain how you made the shifting of the observations?

We have added an explanation in the figure captions. For example: "Observed extreme temperature values are plotted as dots and shown three times—shifted to represent pre-industrial, present-day (2023), and future climates—by subtracting or adding the

product of the global warming index (GWI) and the estimated GWI coefficient from the non-stationary GEV model's location parameter".

- Maybe it is a problem of the generation of the file corresponding to the manuscript, but the figures have some problems of definition for their correct visualisation.

  The conversion of the file to pdf decreased the quality of the figures. We will try to improve it in this new version and/or include the figures in individual files with high resolution.